# Development of Artificial Neural Network-Based Prediction Model for Evaluation of Maxillary Arch Growth in Children with Complete Unilateral Cleft Lip and Palate

**DOI:** 10.3390/diagnostics13193025

**Published:** 2023-09-22

**Authors:** Mohamed Zahoor Ul Huqh, Johari Yap Abdullah, Matheel AL-Rawas, Adam Husein, Wan Muhamad Amir W Ahmad, Nafij Bin Jamayet, Maya Genisa, Mohd Rosli Bin Yahya

**Affiliations:** 1Orthodontic Unit, School of Dental Sciences, Health Campus, Universiti Sains Malaysia, Kubang Kerian, Kota Bharu 16150, Malaysia; dr.mdzahoor@gmail.com; 2Craniofacial Imaging Lab, School of Dental Sciences, Health Campus, Universiti Sains Malaysia, Kubang Kerian, Kota Bharu 16150, Malaysia; 3Prosthodontic Unit, School of Dental Sciences, Health Campus, Universiti Sains Malaysia, Kubang Kerian, Kota Bharu 16150, Malaysia; adamkck@usm.my; 4Department of Biostatistics, School of Dental Sciences, Health Campus, Universiti Sains Malaysia, Kubang Kerian, Kota Bharu 16150, Malaysia; wmamir@usm.my; 5Division of Restorative Dentistry (Prosthodontics), School of Dentistry, International Medical University, Bukit Jalil, Kuala Lumpur 57000, Malaysia; dr.nafij@gmail.com; 6Biomedical Programme, Faculty of Pascasarjana, YARSI University, Jakarta 10510, Indonesia; maya.genisa@yarsi.ac.id; 7Oral & Maxillofacial Department, Hospital Raja Perempuan Zainab II, Kota Bharu 15586, Malaysia; drrosli@yahoo.com

**Keywords:** unilateral cleft lip and palate, artificial neural network, logistic regression, maxillary arch, non-syndromic cleft

## Abstract

Introduction: Cleft lip and palate (CLP) are the most common congenital craniofacial deformities that can cause a variety of dental abnormalities in children. The purpose of this study was to predict the maxillary arch growth and to develop a neural network logistic regression model for both UCLP and non-UCLP individuals. Methods: This study utilizes a novel method incorporating many approaches, such as the bootstrap method, a multi-layer feed-forward neural network, and ordinal logistic regression. A dataset was created based on the following factors: socio-demographic characteristics such as age and gender, as well as cleft type and category of malocclusion associated with the cleft. Training data were used to create a model, whereas testing data were used to validate it. The study is separated into two phases: phase one involves the use of a multilayer neural network and phase two involves the use of an ordinal logistic regression model to analyze the underlying association between cleft and the factors chosen. Results: The findings of the hybrid technique using ordinal logistic regression are discussed, where category acts as both a dependent variable and as the study’s output. The ordinal logistic regression was used to classify the dependent variables into three categories. The suggested technique performs exceptionally well, as evidenced by a Predicted Mean Square Error (PMSE) of 2.03%. Conclusion: The outcome of the study suggests that there is a strong association between gender, age, and cleft. The difference in width and length of the maxillary arch in UCLP is mainly related to the severity of the cleft and facial growth pattern.

## 1. Introduction

Oro facial clefts are the most prevalent facial abnormalities among all ethnicities and ethnic groups. Every day, around 700 infants with cleft lip (CL) and/or cleft palate (CP) are born throughout the world, implying that an oral cleft baby is born every 2 min, or 240,000 children every year [1]. The prevalence of cleft lip and palate (CLP) varies significantly between ethnic groups. CLP is estimated to occur in 1 in 1000 births in whites, 1 in every 500 births for Asians and Native Americans, and 1 in every 2400 to 2500 births in individuals of African ancestry. In Europe, the total incidence of oral cleft is commonly reported to be 1 in 700 live births [2,3,4,5]. Congenital anomalies affect approximately 1 out of every 611 newborns in Malaysia [6]. CL with or without CP is an epidemiologically and etiologically distinct category from isolated CP [7]. CL is related to CP in 68% to 86% of instances [8]. The most frequent congenital malformation among CLP is non-syndromic cleft lip and palate (NSCLP) [2,9]. Unilateral cleft lip and palate (UCLP) is one of the most significant and frequent congenital craniofacial defects, with a high frequency of 6.64 per 10,000 live birth newborns globally [10,11].

Birth prevalence of oral clefts may vary due to sampling type, race, populace, and criteria of inclusion [5]. The presence of a cleft causes apparent craniofacial morphological variations that require subsequent reconstructive surgery of the face after development is complete. Despite the fact that numerous cephalometric analysis studies show that both the maxilla and mandible demonstrate abnormal growth, the focus of current research was essentially on the development of the maxilla rather than comprehensive assessment of the whole mandible and cranial base [12,13].

Dentofacial abnormalities are classified according to the relationship of the maxilla and mandible to each other and to the cranial base. Individuals with a straight facial profile will have normal occlusion and a normal relationship between the maxilla and mandible. This can be classified as Class I. Class II malocclusion frequently comprises mandibular teeth that are too posterior to the maxillary teeth and is combined with a retrognathic mandible; this occurs in around 10% of the population and requires surgical correction in 1%. Class III malocclusion is characterized by mandibular teeth in front of maxillary teeth and is linked with a prognathic mandible, a hypoplastic maxilla, or both [13]. Because of the cleft alveolus, the dentoalveolar segment develops palatal collapse on the afflicted side, resulting in anterior and posterior crossbites. Failure to undergo alveolar bone grafting and orthodontics during the developing phase exacerbates the severity of the malocclusion. Maxillary hypoplasia is a secondary deformity that results from cleft lip and palate surgery, with a reported incidence of 9–45 percent, with isolated cleft lip cases having the lowest incidence rate [14].

Abnormal growth of the maxilla in the sagittal plane is frequently observed in individuals with UCLP. This manifestation has been widely recognized by numerous researchers. However, a consensus has been reached among these experts, acknowledging that the growth and direction of the jaw are significantly impacted by the earlier treatment protocol, including factors such as the timing and techniques employed during primary surgeries [15,16,17,18,19,20,21].

Children with UCLP often face a variety of clinical challenges. Delayed tooth eruption, attrition, occlusal interference, tongue intrusion leading to chewing difficulty, temporomandibular discomfort, periodontal problems, and increased vulnerability to dental caries constitute a few complications. However, despite these concerns, both the affected children and their parents tend to prioritize the surgical correction of cleft abnormalities, often disregarding other oral health issues, including functional disorders such as malocclusion [22,23]. Malocclusion refers to an irregularity that results in the misalignment of teeth, causing both cosmetic and functional issues. This misalignment can have a negative impact on the patient’s physical and emotional well-being, hindering their comfort and overall appearance [24].

The type of cleft and other variables, such as the degree of congenital tissue defect and the intrinsic growth potential of craniofacial tissue, can all impact maxillofacial growth and development [25,26]. There are limited data available on the growth and development of the upper dental arch and occlusion in children with cleft palate. It has been demonstrated that the diameters of the dental arches were reduced in the CP group as compared to the control group [27].

The following methods were used to identify patients who have maxillary constriction (hypoplasia of the maxilla). Individual tooth-to-tooth measurement across the palate and arch width was noted, being that this is one of the factors that impacts the final arch form and is critical in achieving the ideal occlusion. Additionally, Banker et al. demonstrated that the palatal surface of the first molar usually aligned with the cusp tip of the canine in sagittal projections of the normal maxillary arches. They suggested that the transverse dimension of the maxilla can be calculated from the intercanine width (ICW), the interpalatal molar width (IPMW), and their ratio. Given this, the authors attempted to use this proportion in a quantitative description of the arch’s shape [28].

Machine learning, a subset of artificial intelligence, has grown rapidly in recent years, with much of this growth owing to the ever-increasing volume of medical data. The same holds true with artificial neural networks (ANNs), which are both crucial to and distinct from other machine learning algorithms. ANN is an advanced software model whose functioning is influenced by the brain [29,30]. The multilayered connection between input and output may be correctly simulated using relatively simple computer programming code, adding confidence to the ANN’s usefulness utilizing neuralnet version 1.44.2 in R packages. The ANN design is made up of input, hidden, and output levels [31].

It is essential to evaluate the facial growth during the mixed dentition stage due to the potential for enhanced outcomes resulting from early orthopedic and orthodontic interventions. Currently, there is a lack of relevant data pertaining to the facial profile and maxillary arch dimension of individuals diagnosed with cleft lip and palate (CLP) in Malaysia. Hence, the primary purpose of this study was to determine the maxillomandibular relationship with dental characteristics and to develop a neural network logistic regression model to predict the maxillary arch growth in both UCLP and non-UCLP individuals. Another aim was to evaluate the maxillary arch dimensions in children with only CP and/or CL alone.

## 2. Materials and Methods

The 14 distinct dental characteristics of 100 subjects between the ages of 8–16 years were compared using lateral cephalometric radiographs of both cleft and non-cleft individuals. A prior registration of our study was completed in the National Medical Research Registration (NMRR) web portal. Ethical approval was obtained from the Research Ethics Committee, Universiti Sains Malaysia and from the Medical Research and Ethics Committee (MREC), Ministry of Health Malaysia, with approved study protocol code: USM/JEPeM/21080533. The records of patients who visited the Hospital Universiti Sains Malaysia (HUSM) and Hospital Raja Perempuan Zainab II (HRPZ-II) from July 2011–May 2021 were selected. To gain access to patient data from the HUSM and HRPZ II databases, respectively, a prior permission from the concerned hospital directors was sought for research and publication purposes, without revealing any individual patient’s identity. The privacy and confidentiality of all patient data were strictly maintained.

A standardized data collection proforma was designed to incorporate the following components: socio-demographic characteristics such as age and gender, as well as cleft type and category of malocclusion associated with the cleft. A dataset was created based on these factors. The data file has been provided in Appendix A.

This study employs a novel approach incorporating numerous techniques, including bootstrap, multi-layer feed-forward neural network, and ordinal logistic regression using R-syntax. R is an open-source, freely available programming language. Researchers can also contribute to the development of R applications. R provides exceptional support for transforming unstructured data into structured data; consequently, its support for managing unstructured data is also exceptional. R facilitates the development of superior diagrams and charts. This statistical application facilitates activities associated with machine learning, such as regression and artificial neural networking.

During the modeling process, training data are used for model creation, while testing data are used for validation. The research is divided into two phases: phase one involves the use of a multilayer neural network, while phase two involves the use of an ordinal logistic regression model to analyze the underlying relationship between cleft and the selected variables.

### 2.1. Ordinal Logistic Regression

Ordinal logistic regression (OLR) is used to analyze data when the response variable comprises many categories with an inherent rank or order. The dependent variable must be on an ordinal scale for this purpose. The maxilla position is measured on a category scale in this study. Hence, it must be converted into a three-scaled ordinal variable.

The following dichotomous dependent variables are defined in this study as follows:

Class 1 = if the maxilla is in normal position—Normal

Class 2 = if the maxilla is in anterior position—Excess

Class 3 = if the maxilla is in posterior position—Deficient

We can estimate the ordinal regression after converting the maxillary position reading into an ordinal scale. The maximum likelihood method was used to estimate the regression parameter’s value.

### 2.2. Model Equation

The model for ordinal is given by *yi** = *xi*β + ε*i*.

The dependent variable, however, is categorized. Therefore, we must use:


*CX* (*x*) = 1n [P (Y ≤ j|x)P (Y > j|x)] and 1n (∑pr (event)1 − ∑pr (event)) = β0 + β1 X1 + β2 × 2 + β3 × 3 + β4 × 4 +,…,+ βk Xk


### 2.3. Model Summary

The model can be summarized as 1n (P(Y ≤ j|x)1 − P(Y ≤ j|x)) = αj + βi Xk i = 1…k, j = 1,2,…, p − 1, Where α j = threshold or intercept, βi = parameter of the model, and Xi1 = set of factors or independent variables. The equation 1n (*P* (*Y ≤ j|x*)1 − *P* (*Y ≤ j|x*)) = αj + βi Xk is an ordinal logistic model for *k* predictors with the p − 1 levels response variable.

#### The Bootstrap Method

The bootstrap technique was initiated after selecting a sample from the population. It included producing multiple substituting samples and, frequently, resampling to create a pseudo-population. The ordinal regression analysis was performed using the R software version 4.2.2. A detailed step-by-step explanation is provided in Figure 1:

The bootstrap technique was used after the data were prepared. This technique generates a new sample of the same size as the original, wherein each observation is replicated multiple times. Results that do not meet the criteria have been discarded, as outlined by Efron [32] and Efron and Tibshirani [33].

### 2.4. Methodology Building Using R Syntax

#First, import the tidyverse and neuralnet packages.

install.packages(“tidyverse”)

library(tidyverse)

if(!require(tidyverse)){install.packages(“tidyverse”)}

library(tidyverse)

if(!require(neuralnet)){install.packages(“neuralnet”)}

library(neuralnet)

if(!require(dplyr)){install.packages(“dplyr”)}

library(dplyr)

##STEP 1-Dataset for Biometry Modeling Study/##

Input =(“

Age Gender Cleft Categoryc Category

15 0.00 1.00 3.00 Deficient

14 1.00 1.00 3.00 Deficient

16 1.00 1.00 3.00 Deficient

15 1.00 1.00 3.00 Deficient

15 0.00 1.00 3.00 Deficient

15 0.00 1.00 3.00 Deficient

16 1.00 1.00 3.00 Deficient

16 1.00 1.00 3.00 Deficient

14 1.00 1.00 3.00 Deficient

……………………………

…………………………

14 0.00 1.00 3.00 Deficient

15 1.00 1.00 3.00 Deficient

15 0.00 1.00 3.00 Deficient

16 0.00 1.00 3.00 Deficient

15 0.00 1.00 3.00 Deficient

15 0.00 1.00 3.00 Deficient

16 1.00 1.00 1.00 Normal

15 1.00 1.00 3.00 Deficient

15 1.00 1.00 3.00 Deficient “)

data = read.table(textConnection(Input),header = TRUE)

mydata <- rbind.data.frame(data, stringsAsFactors = FALSE)

iboot <- sample(1:nrow(mydata),size = 1000, replace = TRUE)

Bootdata <- mydata[iboot,]

######################################################################

#Part I: MultiLayer MLFFNN

##Install the Neuralnet Package

if(!require(neuralnet)){install.packages(“neuralnet”)}

library(“neuralnet”)

#STEP 2-Determine the Training and Testing of the Dataset

#70% for Training and 30% For Testing

index = sample (1: nrow(data), round (0.70*nrow(data)))

Training <- as.data.frame(data[index,])

Testing <- as.data.frame(data[-index,])

##STEP 3-Plotting the Architecture of MLFFNN Neural Network

nn <- neuralnet(Category~Age + Gender + Cleft, data = Training,

hidden = c (2,2), linear. Output = F, stepmax = 1000000)

plot(nn)

options(warn = −1)

nn1 <- neuralnet(Categoryc~Age + Gender + Cleft, data = Training,

hidden = c(2,2),act.fct = “logistic”,

linear.output = FALSE, stepmax = 1000000)

plot(nn1)

nn1$result.matrix

##STEP 4-Testing the Accuracy of the Model-Predicted Results

##Predicted Results Are Compared To The Actual Results

Temp_test <- subset(Testing, select = c(“Age”,”Gender”,”Cleft”))

head(Temp_test)

nn1.results <- compute(nn1, Temp_test)

##STEP 5-Results

results <- data.frame(actual = Testing$Categoryc,

prediction = nn1.results$net.result)

##STEP 6-Use The Predicted Mean Squared Error NN (MSE-forecasts the Network)

##As a Measure of How Far the Predictions Are From The Real Data

predicted <- compute(nn1,Testing[,1:3])

MSE.net <- sum((Testing$Categoryc − predicted$net.result)^2)/nrow(Testing)

##STEP 7-Printing the Predicted Mean Square Error

MSE.net

###################Neural Network Parameter Output ######################

##STEP 8-Neural Network Parameter Output

library(neuralnet)

nn1 <- neuralnet(Categoryc~Age + Gender + Cleft,data = Training,

hidden = c(2,2),act.fct = “logistic”,

linear.output = FALSE, stepmax = 1000000)

nn1$result.matrix

###########################Model Validation Calculation###################

#####

results <- data.frame(actual = Testing$Categoryc,prediction = nn1.results$net.result)

predicted1 = results$prediction*abs(diff(range(data$Categoryc))) + min(data$Categoryc)

##STEP 9-Print(Predicted)

actual1 = results$actual*abs(diff(range(data$Categoryc))) + min(data$Categoryc)

##STEP 10-Print(Actual1)

deviation = ((actual1-predicted1))

##Print(deviation)

##STEP 11-Mean Absolute Deviance

value = abs(mean(deviation))

print(value)

accuracy_in_percent = (1 − ((value)/100))*100

accuracy_in_percent

#####################Part II: Modeling Ordinal Model####################

###Build Ordinal Logistic Regression Model

if (! require (MASS)){install.packages(“MASS”)}

library(“MASS”)

polr(as.factor(Category)~Age + Gender + Cleft,data = Bootdata,

Hess = TRUE, method = c(“logistic”))

m < -polr(as.factor(Category)~Age + Gender + Cleft,data = Bootdata,

Hess = TRUE, method = c(“logistic”))

summary(m)

## STEP 12-Store Table

(ctable <- coef(summary(m)))

## STEP 13-Calculate and Store p Values

p <- pnorm(abs(ctable[, “t value”]), lower.tail = FALSE)*2

## STEP 14-Combined Table

(ctable <- cbind(ctable, ‘p value‘ = p))

## STEP 15- Odds Ratios

exp(coef(m))

## 3. Results

The study included 100 individuals, 29 men and 21 females in the UCLP group and 14 males and 36 females in the non-UCLP group. The patients’ ages ranged between 8 to 16 years. A total of 72 Malay and 28 Chinese patients from Malaysian ethnicities were recruited into the study, as shown in Table 1. The sample size was calculated using PS software version 3.9.2.

Figure 1 shows the results of the hybrid method using ordinal logistic regression, where category acts as both a dependent variable and as the study’s output. The ordinal logistic regression was utilized to allocate the dependent data according to three classifications. The exceptional performance of the proposed method is indicated by the Predicted Mean Square Error (PMSE) of 2.03%. A low PMSE value reflects the effectiveness of the conducted analysis. In this study, the data are split in the ratio of 70:30, which means that 70% of the data are used for modelling and 30% for testing. The results of the OLR are summarized and the obtained model is derived using the built-in-R-syntax, which is detailed in subheading “The syntax in R for the proposed Hybrid Method”. The model’s outcomes are presented in Table 2. A strong correlation exists between age, gender, and cleft. The values obtained are statistically significant (*p* < 0.25).

### Bootstrap

The bootstrap approach was embedded in the hybrid method to validate the factor (variables). Table 2 summarizes the detailed output. Considering their clinical significance, three variables were included as inputs for model development in this case. These variables are as follows: Age (β1: −0.2346258; *p* < 0.25), Gender (β2: 0.1198695; *p* < 0.25), and Cleft (β3: −1.2477741; *p* < 0.25). The MLFFNN model architecture with three input and three output variables using R-syntax is displayed in Figure 2.

The preciseness of a forecast is determined by comparing actual and predicted values. The dataset was utilised to evaluate the model generated from the training data set. Comparing actual and predicted data, the distance prediction was implemented. It was possible to determine whether or not the produced method was effective by applying the R syntax-based model assessment approach. Using the proposed methodology, Table 3 displays the “actual” and “predicted” values. The disparity between “actual” and “predicted” values was negligible and there were no statistically significant differences observed between the two groups. This demonstrates that the proposed model is preferable, and an equation has been used to determine the variables.

> ##STEP 6-Printing the Predicted Mean Square Error

> MSE.net

[1] 2.033532

> ##STEP 11-Mean Absolute Deviance

> value = abs(mean(deviation))

> print(value)

[1] 2.466827

> accuracy_in_percent = (1 − ((value)/100))*100

> accuracy_in_percent

[1] 97.53317

## 4. Discussion

The use of ML in oral healthcare has demonstrated its usefulness in improving individuals’ oral health by providing clinicians with a tool that allows them to make early decisions to predict oral health status [34].

In machine learning and cognitive science, an artificial neural network (ANN) is a mathematical model that was developed for prediction or classification by imitating the brain and nervous system. Synaptic connections between natural neurons make up the fundamental structure of artificial neural networks (ANNs). The inputs of ANNs correspond to synapses and are weighted according to the intensity of the individual signals. Activation functions are used to determine the outputs of an ANN **[35]**.

Dendrites are the branches of neurons that receive input data acquired from other neurons or from the surrounding environment. The cell body processes the signal before sending it along the axon to the terminal. Receptive neurons along with the neural circuit or functional organs such as muscles may receive the signal and act in response. A single artificial neuron performs the same function. Feature variables are the signals that are used as inputs to perform pattern recognition; they are also known as predictors, input variables, and covariates. The relative significance of each feature variable is taken into account [36].

Various ML algorithms were used to develop the predictive models, as well as to evaluate the performance of the models in predicting children with CLP [36,37,38]. In the previous literature, the most effective application of AI was the evaluation of hypernasality. Three studies categorized the severity of hypernasality and attempted to detect it. The Support vector machine, ANN, and Deep neural network (DNN) classifiers were used to extract speech features as inputs. The detection of hypernasality was more accurate than the severity of hypernasality [37,39,40].

The dental arch dimensions of the maxillary complex appear to be compromised in CLP patients. The interdental width and length of the maxillary arch were significantly altered in patients with CLP during the mixed and permanent dentition stages, compared to the normal group [41,42]. In children with clefts of the soft palate, the maxillary arch dimension in the transverse plane was significantly larger in comparison to individuals with clefts of the hard palate. However, there were no significant differences seen in the upper arch lengths between the groups with clefts involving unilateral CL alone and the control group [27]. Similarly, Rando et al. demonstrated that the upper intercanine distances in the control and UCL groups were substantially greater than in the CP only and UCLP groups [43]. From our study sample, it is revealed that the deficient maxillary growth in most of the cases was noted. The excess and normal growth of the maxilla was also noted in a few cases. This is because our study sample involved the individual CL with isolated cleft palate, as per the criteria for inclusion.

Studies comparing unoperated cleft lip cases to those with cleft lip repair reveal little or no significant variation in maxillary growth. Thus, it has been demonstrated that cheiloplasty has a minimal effect on the development of the maxilla and its dentition. In contrast, palatal surgery has been identified as the primary cause of midface retardation in growth. As a consequence of palatoplasty surgery, scar tissue forms in the sutured areas. This scar tissue impedes the forward and downward translation of the maxilla, which plays a crucial role in normal development [44,45,46]. Maintaining a healthy pedicle of soft tissue is therefore essential. There have also been suggestions to use modified vertical incisions instead of circumvestibular incisions, but this greatly complicates the procedure because of poor access and visibility. Therefore, it is essential to maintain a healthy pedicle of soft tissue [47].

The narrower upper arch was a common feature in individuals with UCLP who underwent surgical correction, and this narrowing of the arch was due to surgical closure. Decreased upper arch width was also seen in untreated people with CLP [48]. It has been suggested that larger clefts often resulted in a shortened sagittal distance between the second deciduous molar and the central incisor on the right side, leading to decreased anteroposterior growth, and clefts of the palate (CLP and CP) were associated with smaller sagittal and transverse maxillary dimensions than clefts of the lip alone [49]. Although atypical sagittal growth of the maxilla is a common symptom of UCLP, all researchers have concluded that the earlier treatment protocol involving the timing and primary surgery techniques has a significant impact on the growth and direction of the jaw [20,42,50].

The reduction in maxillary arch length seen in children in the UCLP group in the current study might be related to the high frequency of hypodontia reported in UCLP patients. Abd Rahman et al. [51] similarly found a 44.9% frequency of hypodontia in Malaysian non-syndromic UCLP children aged about 3 to 12 years. The results of the present study are consistent with a study by Alam et al. [21], who evaluated lateral cephalograms of both cleft and non-cleft individuals and reported a significant decrease in maxillary arch growth leading to maxillary hypoplasia.

This study developed a machine learning model to predict maxillary arch growth in children with and without cleft lip and/or palate (CLP). The model was trained on a dataset of demographic factors, such as age, gender, and cleft type, as well as the category of malocclusion associated with the cleft. The model was then evaluated on a separate dataset and showed excellent performance, with a Predicted Mean Square Error (PMSE) of 2.03%. The PMSE represents the difference between our estimates and the actual results. The combination of variables producing the minimal Mean Square Error (MSE) prediction was designated as the optimal model for MLFNN. The PMSE value obtained in our study suggests a low probability of error. We applied the given approach effectively, and it is quite beneficial for estimating the probability of occurrences. This model is based on exponential equations. The primary goal was to develop, test, and verify a regression modelling method. The main aim of this research was to create and apply techniques in the field of medical statistics by combining the bootstrap procedure with OLR. The variable selection procedure incorporates expert clinical opinion.

The bootstrap method works by first taking the first set of data and making a huge file from it. Second, the bootstrap method prepares a large file replacement sample. Third, the bootstrap method creates statistical samples and saves them. Fourth, the bootstrap method restarts, performing this process repeatedly, sometimes multiple times. The fifth step involves the preparation of the info for the subsequent phase. The R syntax algorithm allows the application to be integrated with the methodological concept. Training and testing data will be kept separate. The R syntax algorithm links the application to the method-based methodological concept.

Following that, the data were submitted to the bootstrap technique. At this stage, 70% of the bootstrap data were labelled as training, while 30% were labelled as testing. The validation dataset was used to validate the model, while the training dataset was used to develop it. A successful model will have a minimal mean square error. Based on the training and testing sets, the formula which was derived from step 2 of the methodology satisfies this requirement. This formula is recommended to reduce PMSE (which requires attaining the lowest PMSE value). The study’s findings resulted in the greatest potential outcomes for the decision-maker.

Because of the combination of statistical formulations, computation using the created R syntax, and the usage of the OLR package, the suggested technique resulted in exceptionally successful linear modelling. Selecting adequate input parameters, preparing the data for linear modelling, and standardising the data are the most difficult tasks.

In our study, advanced statistical tools such as bootstrap and logistic regression were utilized. This is because bootstrap is increasingly becoming a popular alternative method for estimating the parameters and standard errors of logistic regression models. The purpose of applying these statistical methods was to strengthen the analysis of complex data sets and to uncover latent insights that can unleash innovation in a variety of fields.

New technologies with AI-enabled algorithms that scrutinize images acquired by 3D optical intraoral scanners are widely being used for scoring surgical outcomes in Cleft, Lip, and Palate patients [52], and the application of new 3D printed customized orthodontic appliances have been found to be beneficial for newborn infants with craniofacial malformations. The use of flexible virtual appliances fabricated with updated shapes has been shown to be a facilitator of growth in children with CLP [53].

The data in this study were retrospectively derived from secondary data and the study was restricted to Malaysian Chinese and Malaysian Malay children only. In order to generalize the results, multicenter longitudinal studies must be conducted involving multiple ethnicities. More data, both in terms of quantity and variance, will enable a model to perform more precise predictions. These predictive algorithms, such as one which can determine the pathway of cleft formation via the MLFFNN model, and advanced optical scanners with 3D printed devices will be most likely soon become standard practice in the clinical care of individuals with craniofacial abnormalities [34,53].

The uniqueness of this study is the identification of maxillary arch growth among UCLP patients treated at the combined clinic of HUSM and HRPZ II. This study was greatly aided by the use of a hybrid biometry method embedded with logistic regression analyses. In these cases, an advanced statistical method involving bootstrap and OLR by utilizing R-syntax has successfully demonstrated excellent modelling with greater accuracy in the results, and the model correctly predicted the maxillary arch growth in 97.53% of the cases. This method can assist clinicians in making early decisions for enhanced diagnosis and treatment planning.

## 5. Conclusions

A hybrid model using bootstrap and OLR was built and validated thoroughly in this study. The R syntax for this approach was designed to ensure that the researcher completely understood the illustration. According to regression theory, the bigger the R square (or the greater the value), the more exact the developed model. Apart from that, the goodness of fit of a test may be determined using the simplest model for classification or association. Our study demonstrates substantial differences in several characteristics when assessing the maxillary arch dimensions in individuals with UCLP and non-UCLP. The outcome of the study suggests that there is a strong association between gender, age, and cleft. The difference in width and length of the maxillary arch in UCLP is mainly related to severity of the cleft and facial growth pattern. It is concluded that clefts involving the palate (CLP and CP) led to shorter sagittal and transverse maxillary dimensions than clefts affecting only the lip. This information is essential for the clinician when planning an orthodontic treatment, especially during the time of mixed dentition, when accelerated growth favors an effective treatment outcome.

## Figures and Tables

**Figure 1 diagnostics-13-03025-f001:**
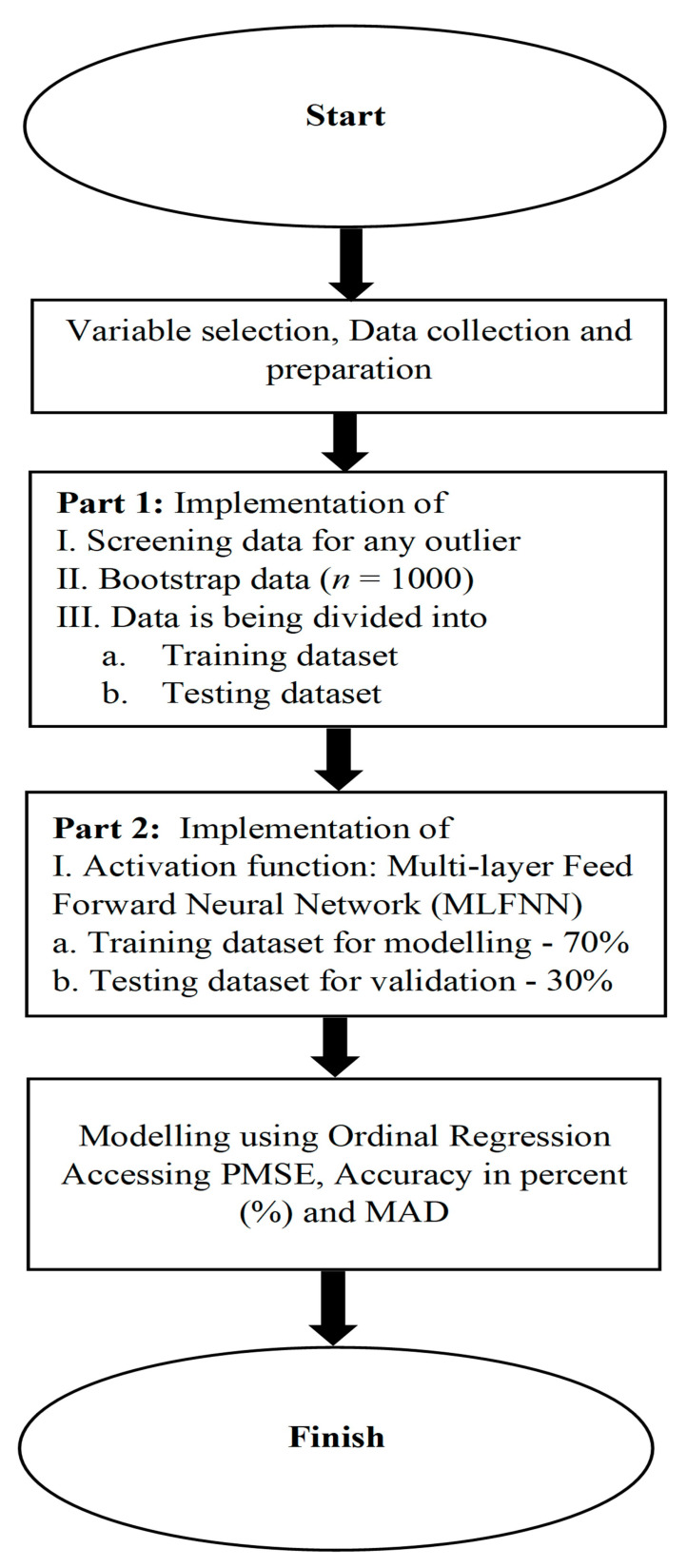
Flowchart of the proposed statistical ordinal modelling is presented to show the technique. This study has a major advantage in incorporating a model that considers clinically significant variables (Predicted mean square error (PMSE) and Mean absolute deviance (MAD)).

**Figure 2 diagnostics-13-03025-f002:**
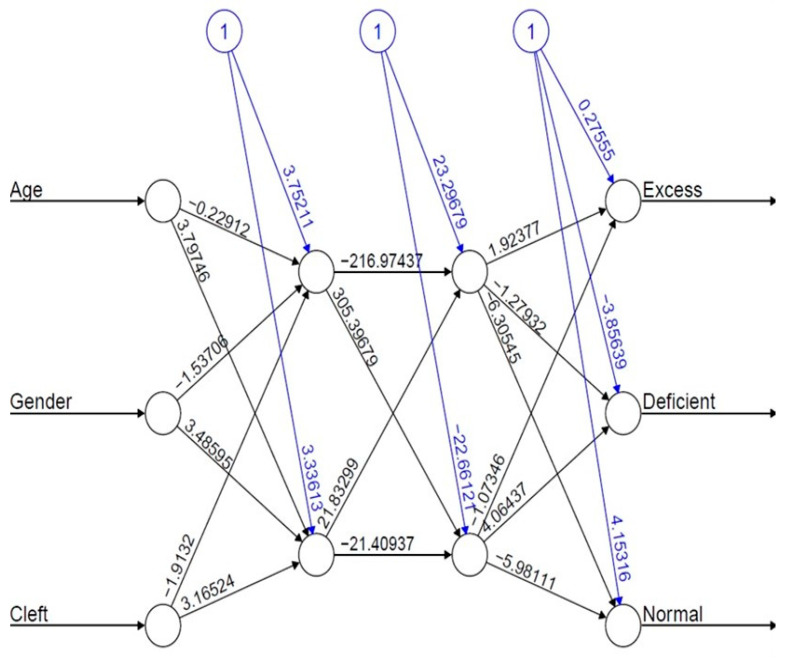
The architecture of the MLFFNN model with three input variables, two hidden layers and three output layers with nodes marked as “1” in blue are known as bias units.

**Table 1 diagnostics-13-03025-t001:** Demographic characteristics.

Variables	UCLP	Non-UCLP	Frequency (%)	
	Mean (SD)	Mean (SD)	UCLP	Non-UCLP
Age	14.56 (1.24)	12.94 (2.30)	-	-
Sex				
Male	-	-	29 (58)	14 (28)
Female	-	-	21 (42)	36 (72)
Ehnic group				
Malay	-	-	39 (78)	33 (66)
Chinese	-	-	11 (22)	17 (34)

SD = Standard deviation, UCLP = Unilateral cleft lip and palate.

**Table 2 diagnostics-13-03025-t002:** Results of Ordered Logistic Regression by Combining the Bootstrap Method.

Variable	Estimate	Std. Error	*t* Value	*p*-Value
Age	−0.2346258	0.0324986	−7.2195676	5.215314 × 10^−13^ *
Gender	0.1198695	0.1287074	0.9313337	3.516810 × 10^−1^ *
Cleft	−1.2477741	0.1464901	−8.5178031	1.626084 × 10^−17^ *
Deficient|Excess	−3.7506324	0.4316391	−8.6892792	3.647463 × 10^−18^ *
Excess|Normal	−1.5794811	0.4104307	−3.8483501	1.189160 × 10^−4^ *

*p* = significant value, *t* value = parameter estimate, Assumption was met, *** Significant at the level of 0.25.

**Table 3 diagnostics-13-03025-t003:** Summary of “Actual” and “Predicted” Values of the Proposed Model.

Actual	Predicted
1	0.996924
0	0.446150
1	0.999999
1	0.998960
0	0.587971
0	0.599176

0 = non-cleft, 1 = cleft.

## Data Availability

The data reported in this study are accessible from the corresponding author upon request. Due to privacy and ethical concerns, the data is not publicly available.

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
