# Peer review of "Development of Artificial Neural Network-Based Prediction Model for Evaluation of Maxillary Arch Growth in Children with Complete Unilateral Cleft Lip and Palate"

_diagnostics, 2023, doi:10.3390/diagnostics13193025_

Round 1

Reviewer 1 Report

1-What is the innovation in this paper?

2-Is your dataset publicly available or private?

3-Why did you not utilize ten-fold cross-validation?

4-Enhance the quality of the figures.

5-Conduct a more comprehensive comparison of your results with other papers in this field.

Minor editing of English language required

Author Response

Comment Number

Original comments of the reviewer

Reply to the comments

Changes done on page number and line number

(Highlighted in Yellow color)

1

What is the innovation in this paper?

Addressed

Thanks to the reviewer for insightful comments. Kindly refer to lines 591-598 of the revised manuscript.

2

Is your dataset publicly available or private?

Addressed

A detailed explanation regarding the privacy and confidentiality has been added. Kindly refer to 135-138 of the revised manuscript.

3

Why did you not utilize ten-fold cross-validation?

Addressed

In our study, the bootstrap method was used to validate the model performance. It works by performing random sampling with replacement from the original dataset and presuming that the unselected data points represent the test dataset. We can repeat this procedure multiple times and calculate the average score to estimate the performance of our model.

Whereas, Cross-validation is performed by dividing the training data into k segments. The k-1 portions are assumed to be the training set, while the other part is the test set. We can repeat this k times while omitting a distinct portion of the data each time. Finally, we use the mean of the k scores to estimate performance. Cross-validation is susceptible to bias and variance. Increasing the number of divisions will increase the variance and decrease the bias. On the other hand, if we decrease the number of divisions, the bias will increase, and the variance will decrease as a result.

4

Enhance the quality of the figures.

Addressed and corrected

The figures have been changed to 300 dpi (Figure 1 and 2)

5

 Conduct a more comprehensive comparison of your results with other papers in this field.

Addressed

Kindly refer citation nos. 44, 51-54 which compared with our study results.

The detailed reply can be found in the attachment.

Reviewer 2 Report

This study developed a machine learning model to predict maxillary arch growth in children with and without cleft lip and/or palate (CLP). The model was trained on a dataset of demographic factors, such as age, gender, and cleft type, as well as the category of malocclusion associated with the cleft. The model was then evaluated on a separate dataset and showed excellent performance, with a Predicted Mean Square Error (PMSE) of 2.03%.

Dear authors, it was a pleasure to read your manuscript. There are some possible improvements I suggest below. Please consider them.

In regard to the 
Materials and Methods section

·         Describe the data collection methods in more detail. 

·         Explain the statistical methods used in the study. Why were these methods chosen?

Regarding the Discussion section

·         Start by summarizing the main findings of the study.  How do your findings compare to previous studies?

·         Discuss the implications of the findings. What do the findings mean for the understanding of maxillary arch growth in CLP children? What are the clinical implications of the findings?

·         Identify the limitations of the study. What are the factors that could have influenced the results of the study? What are the areas for future research?

·         Strengthen writing by using more specific language. For example, instead of writing "The model has shown the accuracy of 97.53%," write "The model correctly predicted the maxillary arch growth in 97.53% of the cases."

·         Use more consistent terminology. For example, throughout the discussion section, the authors use different terms to refer to the same thing. For example, they refer to the bootstrap method as "bootstrapping" in one sentence and "bootstrap" in another sentence.

·         Avoid using jargon and technical terms that may not be familiar to the reader. For example, the authors use the term "PMSE" without explaining what it means.

·         Discuss whether similar AI approach can be applied in the near or distant future for prediction of growth in patients with other craniofacial disorders- For example for support of personalized device designs –as a support to adapt their personalized appliances better. Consider referencing recent study on the use of optical scanning and 3D printing to fabricate customized appliances for patients with craniofacial disorders - DOI:https://doi.org/10.1053/j.sodo.2022.10.005 as well as discuss if non-invasive 3D facial scans can be used to analyze the underlying association between cleft and facial proportions.

I hope these suggestions were helpful.

Is fine.

Author Response

Comment Number

Original comments of the Reviewer

Reply to the comments

Changes done on page number and line number

(Highlighted in red color)

1

Describe the data collection methods in more detail

Addressed and corrected

Thanks to the reviewer for insightful comments. Kindly refer to lines 129 – 138 of the revised manuscript.

2

Explain the statistical methods used in the study. Why were these methods chosen?

Addressed

Thanks to the reviewer for insightful comments. Kindly refer to lines 569 - 574 of the revised manuscript.

3

Start by summarizing the main findings of the study.  How do your findings compare to previous studies?

Addressed

Kindly refer to citations of reference nos. 28,42,43 and 44 of the discussion part of the revised manuscript. .

4

Discuss the implications of the findings. What do the findings mean for the understanding of maxillary arch growth in CLP children? What are the clinical implications of the findings?

Addressed

Kindly refer to lines 528 - 534 of the revised manuscript.

5.

Identify the limitations of the study. What are the factors that could have influenced the results of the study? What are the areas for future research?

Addressed

Kindly refer to lines 583 - 590 of the revised manuscript.

6

 Strengthen writing by using more specific language. For example, instead of writing "The model has shown the accuracy of 97.53%," write "The model correctly predicted the maxillary arch growth in 97.53% of the cases."

Addressed and corrected

The statement has been added. Kindly refer to lines 596-598 of the revised manuscript.

7

Use more consistent terminology. For example, throughout the discussion section, the authors use different terms to refer to the same thing. For example, they refer to the bootstrap method as "bootstrapping" in one sentence and "bootstrap" in another sentence.

Addressed and corrected

The sentence bootstrap has been added all over the manuscript.

The detailed reply can be found in the attachment.

Round 2

Reviewer 1 Report

thanks for your answers.

Reviewer 2 Report

OK

OK